# Receptor Tyrosine Kinase Inhibitors for the Treatment of Recurrent and Unresectable Bone Sarcomas

**DOI:** 10.3390/ijms232213784

**Published:** 2022-11-09

**Authors:** Víctor Albarrán, María Luisa Villamayor, Jesús Chamorro, Diana Isabel Rosero, Javier Pozas, María San Román, Juan Carlos Calvo, Patricia Pérez de Aguado, Jaime Moreno, Patricia Guerrero, Carlos González, Coral García de Quevedo, Pablo Álvarez-Ballesteros, María Ángeles Vaz

**Affiliations:** Department of Medical Oncology, Ramon y Cajal University Hospital, 28034 Madrid, Spain

**Keywords:** tyrosine kinase receptors, TKI, target, osteosarcoma, Ewing sarcoma, chondrosarcoma, chordoma

## Abstract

Bone sarcomas are a heterogeneous group of rare tumors with a predominance in the young population. Few options of systemic treatment are available once they become unresectable and resistant to conventional chemotherapy. A better knowledge of the key role that tyrosine kinase receptors (VEGFR, RET, MET, AXL, PDGFR, KIT, FGFR, IGF-1R) may play in the pathogenesis of these tumors has led to the development of multi-target inhibitors (TKIs) that are progressively being incorporated into our therapeutic arsenal. Osteosarcoma (OS) is the most frequent primary bone tumor and several TKIs have demonstrated clinical benefit in phase II clinical trials (cabozantinib, regorafenib, apatinib, sorafenib, and lenvatinib). Although the development of TKIs for other primary bone tumors is less advanced, preclinical data and early trials have begun to show their potential benefit in advanced Ewing sarcoma (ES) and rarer bone tumors (chondrosarcoma, chordoma, giant cell tumor of bone, and undifferentiated pleomorphic sarcoma). Previous reviews have mainly provided information on TKIs for OS and ES. We aim to summarize the existing knowledge regarding the use of TKIs in all bone sarcomas including the most recent studies as well as the potential synergistic effects of their combination with other systemic therapies.

## 1. Bone Sarcomas: Introduction and Background

### 1.1. Epidemiology

Sarcomas are malignant tumors of mesenchymal origin with a low incidence in the general population, accounting for less than 1% of all cancer diagnoses. More than 70 histologic subtypes have been identified, although they can be broadly classified into two general subgroups: soft tissue sarcomas and sarcomas of the bone.

According to the EUROCARE (European Cancer Registry-based study on survival and care of cancer patients) database [1], bone sarcomas make up <0.2% of malignant tumors, with distinct patterns of incidence for different histologic subtypes. Osteosarcoma (OS) is the most common primary bone tumor overall and has a higher incidence in the second decade of life (0.8–1.1/100,000 per year at age 15–19 years). Similarly, Ewing sarcoma (ES) is relatively common in adolescents and young adults, whereas chondrosarcoma (CS) is more frequent in older adults [2]. Less frequent subtypes include chordomas, giant cell tumors of bone (GCTB), pleomorphic sarcomas, and dedifferentiated or poorly differentiated bone tumors.

Persistent and non-mechanical bone pain, usually accompanied by progressive swelling and functional impairment, is the most common pattern of clinical presentation and should prompt a quick radiological assessment. The work-up and therapeutic management of bone sarcomas should be carried out by expert multidisciplinary teams at reference centers [3].

### 1.2. Principles of Treatment

Low-grade localized OS should be treated by surgery alone, given its low metastatic potential, whereas high-grade OS usually requires a period of preoperative chemotherapy (CT). The MAP regimen (doxorubicin, cisplatin, and high-dose methotrexate) is usually used as front-line treatment for young patients (<40 years).

The management of recurrent or metastatic OS should follow the same principles as the localized OS when the lesions are resectable. Retrospective data (EURAMOS-1) suggest that metastatic high-grade OS may have a similar prognosis to that of localized disease when the surgical removal of all known lesions is achievable, with a 3-year and 5-year event-free survival of 59% (95% confidence interval [CI]: 57–61%) and 54% (95% CI: 52–56%), respectively. Multivariate analyses suggest that pulmonary metastases and an axial skeleton tumor site are the most adverse prognostic factors at diagnosis (hazard ratio [HR] 2.34 and 1.53) [4].

However, the treatment of metastatic unresectable OS remains challenging, with a 5-year post-relapse survival rate that remains <20% despite systemic treatment [5]. Second-line conventional CT includes ifosfamide or cyclophosphamide in association with etoposide or carboplatin (III, B) and other active drugs with a lower grade of evidence such as gemcitabine and docetaxel (IV, C) [6].

In the case of ES, a multimodal approach is recommended, with an overall 10 to 12-month long treatment of induction CT, local therapy, and consolidation CT thereafter. CT regimens include vincristine (V), doxorubicin (D), cyclophosphamide (C), ifosfamide (I), and etoposide (E), usually using interval compressed VDC/IE [7]. Metastatic presentation (occurring in 25% of patients) is the most important prognostic factor, with a 5-year survival of 60–75% in localized disease and 20–40% in metastatic disease [8].

According to preliminary results from the rEECur study, topotecan plus cyclophosphamide and high-dose ifosfamide may be the preferred regimens for further lines of CT, followed by temozolomide plus irinotecan and gemcitabine plus docetaxel [9]. Nevertheless, these regimens are not standardized and have low response rates, so the prognosis of recurrent ES remains very poor.

Inoperable locally advanced or metastatic high-grade chondrosarcomas have a poor prognosis and CT is of limited benefit [10]. Similarly, systemic CT is generally inactive and not recommended in rarer bone tumors such as chordomas [11], denosumab-refractory GCTB [12], or undifferentiated pleomorphic sarcomas [13].

Therefore, advanced unresectable bone sarcomas remain orphan diseases in which there is an urgent need to find new active drugs that improve the prognosis of patients. In recent years, a better understanding of the oncogenesis and molecular pathways that orchestrate the biological behavior of these tumors has prompted the research of new targeted therapies that may soon be part of our therapeutic arsenal.

The use of drugs that simultaneously block several tyrosine-kinase receptors (TKRs) has emerged as a promising strategy to improve the results of systemic treatment. The purpose of this review was to summarize the molecular basis, the biological rationale, and the clinical results of tyrosine-kinase inhibitors (TKIs) in patients with advanced bone sarcomas.

## 2. TKIs in Bone Sarcomas: Biological Rationale

### 2.1. Osteosarcoma

The occurrence of OS seems to be related to an aberration in the physiologic process of bone growth and remodeling. The incidence of OS is higher during the periods of rapid bone growth and the most common sites of origin are the metaphysis of long bones, precisely those with the greatest increase in bone length. This may be explained by a predisposition of rapidly proliferating cells to accumulate mitotic mistakes and other genetic events that may ultimately lead to oncogenic transformation [14].

In contrast to other sarcomas, the oncogenesis of OS is not driven by a characteristic translocation, although recurrent deletions and amplifications have been identified in multiple chromosomal regions. According to recent whole-genome sequencing (WGS) studies, nearly 33% of OS show massive genomic rearrangements and chromosome remodeling, which may arise from a single genetic catastrophe (a phenomenon known as *chromothripsis*), whereas in other solid tumors, only 2–3% show such features [15]. This might explain the ‘chromosomal chaos’ and highly complex karyotypes that characterize OS.

These alterations interfere with the differentiation of mesenchymal stem cells (MSCs) into mature osteoblasts, giving rise to immature ‘preosteoblasts’. These cells show a higher capacity for self-renewal and an increased secretion of osteoclast-stimulating cytokines that favor bone resorption and osteolysis, disturbing the homeostasis of the bone microenvironment [16].

The most frequent deletions in OS are those affecting the regions 3q, 13q, and 17p, leading to genomic alterations in the *RB* and *TP53* genes [17]. The most common amplifications involve the genes *MYC* (whose amplification has been associated with a poor prognosis), *FGFR1*, and *IGF1R*. Around 25% of patients have alterations in the phosphatidylinositol 3-kinase (PI3K)/mTOR pathway, mainly mutations or loss of *PTEN* [18,19]. Molecular alterations have also been reported on *DLG2*, *ATRX*, and *CD4* [15].

More interestingly for a potential therapeutic approach, an overexpression of multiple tyrosine-kinase receptors (TKRs) has been detected in OS. When dimerization of TKRs is induced by the binding of an extracellular ligand, they acquire the capacity to phosphorylate tyrosine residues in specific substrates. This stimulates the activation of downstream intracellular signaling pathways that are related to cell differentiation, proliferation, angiogenesis, and control of the cell cycle, amongst other essential processes [20].

The abnormal activation of TKRs can be unleashed by different mechanisms (mutation, enhanced expression or autocrine stimulation), ultimately resulting in a constitutive activity that facilitates oncogenesis through the progressive acquisition of cancer biological hallmarks.

Although VEGFR and RET are currently the most relevant TKRs in OS, several other molecules (PDGFR, FGFR, KIT, MET, IGF-1R, AXL) might play a role as potential targets in the future. The most important biological effects and downstream pathways activated by TKRs are summarized in Figure 1.

VEGFR is a family of TK receptors for vascular endothelial growth factor (VEGF), with three main isoforms (VEGFR1, VEGFR2, and VEGFR3), with VEGFR2 the most relevant for the activation of intracellular signaling intermediates and the promotion of angiogenesis [21]. According to different studies, a high expression of VEGF is associated with a lower progression-free and overall survival in patients with OS [22]. However, the use of monoclonal antibodies that inhibit just VEGF (bevacizumab) has shown disappointing results in OS patients [23], suggesting that the efficacy of inhibiting just VEGF/VEGFR is limited and should be combined with the blockade of other pathways.

The VEGFR family is aa common target of many TKIs (imatinib, lenvatinib, regorafenib, axitinib, cabozantinib, cediranib, apatinib, sorafenib), some of which show specificity for certain isoforms. Imatinib is the most powerful inhibitor of all three isoforms. Apatinib and cabozantinib preferentially block VEGFR1, whereas regorafenib and sorafenib have a stronger effect on VEGFR3 [24,25].

RET is a transmembrane receptor that forms a heterodimeric complex with the glial cell line-derived neurotrophic factor (GDNF) family ligands, leading to the activation of pro-oncogenic signaling pathways such as RAS/mitogen activated protein kinase (MAPK) and PI3K/AKT. This is of vital importance for the development of neural and neuroendocrine tissues as well as morphogenesis and stem cell maintenance [26].

Though its role in OS has been far less studied than in other malignancies, it is known that RET can promote the oncogenic behavior of metastatic OS [27]. The stem cell-like properties in OS cells seem to be facilitated by the overexpression of RET, which has been associated with a poor prognosis and a higher resistance to different chemotherapeutic agents [28,29]. Though not studied in detail thus far, RET might be a potentially critical target for OS treatment.

The platelet-derived growth factor (PDGF) receptors (PDGFR-α and PDGFR-β) are also TKRs with a potential relevance in OS pathogenesis. Though its expression is heterogeneous and its correlation with prognosis has not been clearly established, there is some evidence implicating the PDGFs/PDGFRs pathway in the oncogenesis of OS cells.

A study of 35 samples of human OS found positive immunohistochemistry staining of hypoxia marker hypoxia-inducible factor 1α (HIF-1α), suggesting that hypoxia is a significant feature of OS cells [30]. Hypoxia seems to upregulate the transcription and expression of PDGF and PDGFR, which are associated with enhanced cell migration and proliferation through the stimulation of the AKT/ERK and STAT3 signaling pathways [31].

Fibroblast growth factor receptors (FGFR) are a family of TKRs with a strong involvement in many physiological events related to inflammation and tissue repair, whose aberrant activation due to mutations, amplifications or gene fusions has been detected in many solid tumors. FGFR amplification has been associated with OS resistance to chemotherapy [32] and the development of lung metastases [33].

Though less investigated to date, other TKRs such as KIT [34], MET [35], IGF-1R [36] and AXL [37] are overexpressed and implicated in the progression of OS, and are associated with a poorer prognosis and a lower response rate to systemic treatment.

### 2.2. Ewing Sarcoma

The key event in the tumorigenesis of ES is the translocation t(11;22)(q24;q12) of the N-terminus of *EWSR1* to the C-terminus of *FLI1*, resulting in the genesis of a EWS/FLI1 fusion protein that binds RNA helicase A and directly interacts with the gene promoters [38]. This represents a unique mechanism of regulation affecting essential enhancer elements for cell proliferation and oncogenesis [39]. Several studies have associated this oncogenic event with the ES cell enrichment with proteins that favor the dissemination of aberrant cells (MMP2, MMP9, MT1-MMP) [40]. The inactivation of *TP53* and *CDKN2A* genes also seems to play a role in the dysregulation of key physiological processes that allows oncogenesis in ES [41].

Though few signaling pathways have emerged as therapeutic targets and the clinical development of TKIs is weaker than in OS, angiogenesis and other processes regulated by TKRs begin to gain importance as prognostic factors and hallmarks of ES oncogenesis.

More than 90% of ES show an upregulation of insulin-like growth factor I receptor (IGF-1R) [42], which seems to promote the transcriptional expression of EWS fusion genes and facilitate several oncogenic pathways [43]. Its expression has been associated with a worse clinical prognosis [44] and a higher resistance to chemotherapy, since the IGF-1R mediated activation of the PI3K/AKT pathway prevents the apoptosis of ES cells induced by the cytotoxic effect of some anticancer drugs [45].

Although the importance of VEGFR is not as established as in OS, some VEGFR isoforms can increase the tumor vessel density in ES, facilitating tumor growth and cell proliferation [46]. More importantly, the expression of VEGF in ES cells also seems to be induced by IGF-mediated stimulation of the MAPK and PI3K/AKT signaling pathways [47], making the dual blockade of VEGFR/IGFR an interesting therapeutic approach.

As observed in OS, FGFR is highly expressed in ES cells. In particular, FGFR1 is highly activated in nearly 80% of ES samples, probably promoting tumor progression and dissemination by activating the PI3K/AKT, RAS/MAPK, and JAK/STAT signaling pathways [48]. Interestingly, the EWS/FLI1 fusion protein can significantly reduce the expression of SPRY1, a negative regulator of the FGFR-activated MAPK pathway, thereby suppressing one of the main negative-feedback mechanisms to downregulate its oncogenic effects [49]. Several activating mutations in different *FGFR* isoforms such as *FGFR1 N546K* [50], *FGFR3 K650E* [51], and *FGFR4 G388A* [52] have been reported in ES samples as possible enhancers of tumor aggressiveness in comparison with the *FGFR* wild-type tumors.

Human epidermal growth factor receptors (ERBBs) have also been identified as contributors to the survival and progression of ES cells through the activation of downstream pathways. Particularly, around 15% of ES patients have a HER2 upregulation, which seems to reduce the survival rate and confer resistance against anti-topoisomerase I anticancer drugs such as adriamycin and etoposide [53]. The expression of HER3 and HER4 is increased in ES cell lines and is associated with shorter survival rates, possibly due to an increase in the Rac1 GTPase activity, related to the activation of the PI3K/AKT and FAK pathways [54]. The expression of epidermal growth factor receptor ERBB1 (EGFR) is generally low in ES cell lines, and its possible contribution to ES oncogenesis remains unknown.

PDGFR-α and PDGFR-β are also highly expressed in both ES cells and in the endothelial and pericyte-like cells of the tumor microenvironment. PDGFR seems to favor the motility of ES cells through the activation of the PI3K/AKT and phospholipase C gamma (PLC-γ) downstream pathways [55]. According to studies in preclinical models, the knockdown of PDGFR-β in ES cells significantly reduces the risk of lung metastasis [56], suggesting its role in the dissemination of ES cells.

Other TKR with a possible implication in ES oncogenesis are AXL [57], MET [58], KIT [59], and interferon-α/β receptor (IFNAR) [60], though further research is required to establish their clinical relevance.

### 2.3. Other Bone Sarcomas

Although the importance of TKRs as therapeutic targets is mainly established in OS and ES, research has also begun to identify their potential role in the pathogenesis of other primary bone tumors.

A preclinical study detected the strong phosphorylation of S6 kinase, a surrogate of the PI3K/AKT pathway activity, in 69% of conventional and 44% of dedifferentiated chondrosarcoma (CS) clinical samples, and showed that several TKIs suppressed S6 phosphorylation in vitro [61]. The constitutive activation of PI3K/AKT may be mediated by the overexpression of PDGFR and IGF-R1, which has been demonstrated in CS cell lines [62,63].

In addition, there is growing evidence on the importance of angiogenesis in the progression of CS and the increase in aberrant vascularization in higher histological grades [64]. Preclinical studies suggest that TKR inhibitors targeting PDGFR, VEGFR, and FGFR are able to inhibit the growth of CS in animal models due to their antiangiogenic effects [65].

In chordomas, the overexpression of EGFR has been widely reported. Shalaby et al. [66] evaluated the EGFR status by immunohistochemistry in 173 chordoma samples, finding a positive expression in 69%. A study of 22 naïve chordomas found a HER2/neu co-expression in 50% of the cases, without mutations in the corresponding genes [67].

The overexpression of PDGFR and MET has also been identified in chordomas [68,69], with a downstream hyperactivation of the PI3K/AKT pathway, which seems to play an important role in their pathogenesis [70]. Moreover, the blockade of FGFR seems to reduce MEK/ERK phosphorylation, increasing apoptosis and restricting the growth rate of chordoma cell lines [71].

In GCTB, VEGFR has been identified as an enhancer of angiogenesis and osteoclastogenesis, induced by the receptor-activator of nuclear factor kB (RANK) and its ligand (RANKL) [72]. Interestingly, relapsed GCTB after treatment with denosumab seemed to exhibit specific profiles of phosphorylated kinases, with a statistically significant increase in the phosphorylation of EGFR, IGF-1R, and PDGFR-β in comparison with the treatment-naïve tumors [73].

A recent study of 176 samples of undifferentiated pleomorphic sarcoma (UPS) through RNA-sequencing techniques identified two main subgroups of tumors, one with a high density of CD8-positive cells and strongly enriched in genes related to immunity, and a second subgroup enriched with genes involved in the development and progression of stem-like cells [74]. Overexpression of FGFR2 was identified in this ‘immune-low’ subgroup and FGFR inhibitors showed anti-tumor activity in these cell lines in vitro, though deeper research is needed to consider these findings as a biological rationale for the use of TKIs.

## 3. TKIs in Bone Sarcomas: Clinical Results

Considering the contribution of TKRs to the oncogenesis and microenvironment modulation of bone sarcomas, several TKIs have been the subject of preclinical studies and early-phase clinical trials. Since some bone tumors tend to calcify in response to treatment, without an observable shrinkage in their size, progression-free survival (PFS)—and not response rate—is commonly adopted as the primary clinical endpoint. The reported phase II trials of TKIs in bone sarcomas are summarized in Table 1.

### 3.1. Osteosarcoma

OS is the primary bone tumor with the highest evidence for the use of TKIs, mainly regorafenib and cabozantinib. Regorafenib is an oral multi-target TKI that inhibits VEGFR1/2/3, PDGFR, KIT, FGFR-1, and MET [91]. Preclinical studies show that regorafenib upregulates caspase-3 and caspase-8 cleavage, therefore suppressing OS cell growth by prompting apoptosis, and downregulates several genes related to invasion (VEGF, MMP-9) and proliferation (Cyclin-D1) [92].

Two multi-center randomized phase II clinical trials have compared regorafenib to placebo in patients with recurrent or refractory OS, both achieving a significant increase in PFS. The French study REGOBONE included 38 patients and showed a median PFS of 16 weeks (vs. 4 weeks in the placebo group), with a partial response (PR) rate of 7.6% [75]. The North American study SARC024 included 22 patients and achieved a median PFS of 3.6 months (vs. 1.7 months), with a response rate of 13.6% [77]. No significant impact was observed on overall survival (OS).

Cabozantinib is a TKI that targets VEGFR1/2/3, PDGFR, KIT, MET, RET, and AXL and has also a demonstrated an antitumor effect against OS [25]. Preclinical studies show that cabozantinib decreases the migration and proliferation of OS cells through the inhibition of the ERK/AKT pathway and the modulation of the tumor microenvironment [93].

A single-arm phase II clinical trial (CABONE) tested cabozantinib in 42 patients with relapsed OS, reaching a median PFS of 6.7 months, a response rate of 12%, and inhibiting tumor progression at 6 months in one-third of the patients [79]. A retrospective real-world study from the Hellenic Group of Sarcoma and Rare Cancers analyzed clinical data from nine adult patients with advanced OS (*n*: 7) and ES (*n*: 2), who received cabozantinib off-label between April 2019 and May 2020, with a PFS varying from 1 to 8 months and disease stabilization in five patients (56%) [94].

Apatinib is a VEGFR1/2, RET, and KIT inhibitor that seems to promote apoptosis and autophagy, downregulating invasion, migration, and PDL1 expression in OS cells [95]. A single-arm phase II clinical trial evaluated the efficacy of apatinib in 37 patients with advanced OS after the failure of standard multimodal therapy, obtaining an objective response rate of 43% and a median PFS of 4.5 months [96]. The use of a high dose of apatinib (750 mg/day) may explain the good results of the clinical trial, with a higher response rate than that reported by previous retrospective studies with apatinib at 500 mg/day (26% and 33%) [97,98].

Sorafenib and lenvatinib are similar multi-target TKIs with a known effect against OS, whose targets include VEGFR2/3, PDGFR, KIT, FGFR-1, and RET [99,100]. According to preclinical studies, sorafenib blocks angiogenesis, cell growth, and the metastatic dissemination of OS cells through the blockade of the ERK1/2, MCL-1, and ezrin pathways [101]. Thirty-five patients with refractory OS were treated with sorafenib in a non-randomized phase II clinical trial, with three PR and 14 stable diseases (SD), some of them with minor responses (<30% tumor shrinkage) or a reduction in the tumor density observed in FDG-PET [80]. A phase I/II study of single-agent lenvatinib in 31 children and young adults with OS has reported a response rate of 7% and a median PFS of 3 months [82].

Other TKIs have shown limited efficacy in OS. Axitinib and cediranib are drugs with overlapping targets (VEGFR1/2/3, PDGFR, FGFR, and KIT) [102,103] without any preclinical evidence for their use in OS. A phase I clinical trial evaluating axitinib in 19 adolescents with refractory solid tumors included two patients with OS who reached stable disease as the best response [104]. A similar phase I multi-tumor clinical trial with cediranib in 16 patients included four patients with OS, with only one minor response [105].

Imatinib is a KIT, PDGFR, and CSF-1R inhibitor [106] with some activity against OS cells in preclinical murine models [107]. However, a phase II clinical trial in 185 patients with ten different histologic subtypes of sarcomas including 27 patients with OS showed suboptimal results, with only five stable diseases and no objective response [84].

Pazopanib is a multi-TKI that targets VEGFR1/2/3, PDGFR, and FGFR, and is more widely used in soft tissue sarcomas. A multi-case communication of 15 patients with OS reported one PR and eight SD [108], though the only prospective clinical trial with pazopanib (NCT01956669) has not published its results.

### 3.2. Ewing Sarcoma

The evidence for the use of TKIs in ES is more limited than in OS, with only cabozantinib and regorafenib having demonstrated clear antitumor activity in phase II clinical trials. The above-mentioned phase II study with cabozantinib for bone sarcomas (CABONE) included 39 assessable patients with advanced ES, obtaining 10 partial responses (26%) and a median PFS of 4.4 months [79].

The randomized placebo-controlled phase II trial REGOBONE included a cohort of 41 ES patients (36 efficacy-evaluable), reaching five PR (21.7%) and a median PFS of 11.4 weeks (versus 3.9 weeks in the placebo group) [76]. Regorafenib has also been evaluated in a single-arm phase II study (REGO) in 30 patients with advanced ES and Ewing-like tumors, reporting three PR, 18 SD, and a median PFS of 3.6 months [78]. *EWSR1* translocation was detected by FISH in two of the three patients with objective responses.

The above-mentioned phase I clinical trial of cediranib in 16 children and adolescents with refractory solid tumors included three patients with ES, with one PR (reporting a 77% reduction in tumor size) [105]. The phase I trial evaluating axitinib in 19 children and adolescents with refractory tumors included one ES that achieved SD as the best response [104].

A retrospective study of sorafenib in patients with refractory bone tumors included two patients with metastatic ES, one of them achieving a PR [109]. Some retrospective data also showed a modest activity of apatinib in advanced ES, alone [110] or in combination with everolimus [111], though there was no evidence from clinical trials reported up to date.

According to preclinical studies, imatinib could induce dose-dependent apoptosis and inhibit the proliferation of ES cells in vitro and in vivo [112]. A small clinical trial evaluated the efficacy of imatinib in eight patients with recurrent ES-family tumors and desmoplastic small round cell tumors expressing KIT and/or PDGFRα [113]. A partial response was reported in a patient with an intense expression of both markers (3+/4+), giving rise to the hypothesis of the potential use of molecular and/or immunohistochemical factors to predict response.

### 3.3. Other Bone Sarcomas

The low incidence of other bone sarcomas, their clinical heterogeneity, and the consequent rarity of unresectable and disseminated status that require a systemic treatment makes it difficult to conduct clinical trials and generate scientific evidence for the use of TKIs.

Dasatinib, a small molecule inhibitor of the Src family of kinases, PDGFRα/β, and c-KIT, is currently the only TKI that has demonstrated activity against both chondrosarcoma (CS) and chordoma in a phase II clinical trial. This study included 116 patients (33 CS, 32 chordomas, 25 solitary fibrous tumors, 12 alveolar soft-part sarcomas, and seven epithelioid sarcomas), with a median PFS of 5.5 months for CS and 6.3 months for chordoma [87]. An objective response was reported in six patients with CS and six patients with chordoma, with three patients in each group remaining on treatment for more than two years.

A phase II clinical trial confirmed the antitumor activity of imatinib in 56 patients with advanced chordomas. Among the 50 patients evaluable by RECIST criteria [114], one PR and 35 SD—with a minor response of <20% in nine cases—were reported, with a clinical benefit rate of 64% and a median PFS of 9.2 months [88].

Lapatinib is a TKI active against both EGFR and HER2/neu [115]. A phase II clinical trial studied lapatinib in 18 patients with advanced progressing chordoma, with the expression and activation of EGFR evaluated by immunohistochemistry and/or phosphoarrays, reporting six PR (33.3%) and seven SD (38.9%) according to the Choi criteria, with a median PFS of 6 months [90].

The evidence for other agents is weaker. A real-world experience study retrospectively analyzed 44 patients with advanced bone sarcomas treated with TKIs including nine CS [116]. Seven patients were treated with cabozantinib, with four SD and no objective response. Two patients were treated with regorafenib, with one PR and one SD. These data suggest that these TKIs have a lower activity in CS than they have in OS and ES.

A retrospective study of 33 patients treated with apatinib for advanced CS (20 conventional CS, five dedifferentiated CS, four mesenchymal CS, three extraskeletal myxoid CS, and one clear cell CS) reported an objective response in six patients (18.2%), with a median PFS of 12.4 months [117]. These results suggest a meaningful activity of apatinib against high-grade CS, though the significant rate of G3-4 adverse events (33%) should be weighed against the slow growth pattern and relatively indolent behavior of CS.

Anlotinib (multi-target TKI that inhibits VEGFR, FGFR, PDGFR, KIT, and RET) was evaluated by a multiple-institution retrospective study in 48 patients with advanced sarcomas including 27 OS, eight ES, nine CS, and three chordomas [118]. The median PFS varied among pathological subtypes: 4.7 months for OS, 6.7 months for ES, 4.2 months for CS, and 3.2 months for chordoma, with a global ORR of 10.4%, suggesting a discrete activity against different types of bone sarcoma.

A multi-center phase II clinical trial of sunitinib for advanced non-GIST sarcomas included nine chordomas and one malignant giant cell tumor of bone (GCTB), with a durable PR in four chordomas and the patient with GCTB, which achieved disease control for at least 16 weeks [119]. Li et al. [120] reported the case of a patient with a multicentric GCTB of the pelvis and spine who was treated with apatinib and zoledronic acid and experienced a quick symptomatic improvement and a prolonged partial response based on RECIST criteria.

A phase I clinical trial of pazopanib and lapatinib combination therapy in 75 patients with advanced solid tumors included one PR in a patient with GCTB [121]. A translational study investigated the combination of denosumab with lenvatinib in biological samples of GCTB and desmoplastic fibromas, finding a higher antitumor activity compared to denosumab alone [72].

### 3.4. Optimizing the Use of TKIs in Bone Sarcomas

As described above in detail, despite the encouraging results from preclinical studies with TKIs in bone sarcomas, phase I/II single-agent clinical trials have shown at best partial responses or disease stabilization for a few months. This suggests that bone sarcomas have either primary resistance to TKIs or the capacity to develop acquired resistance in a short period of time. The mechanisms of resistance may include the mutation or downregulation of the TKRs, the alteration of downstream effector molecules, or the development of compensatory signaling pathways [122,123], making the TKR inhibition futile.

OS patients from the CABONE trial with high levels of soluble MET and low levels of VEGFA seemed to have a better response to cabozantinib [79], leading to the hypothesis that the measurement of serum proteins related to the target TKR may be used to identify sensitive or resistant patients. The imprecise knowledge of the mechanisms of response and resistance is currently a limitation for the use of TKIs, and further research is required to explore the sequential administration of different TKIs once resistance has developed.

Strategies to overcome the resistance to TKIs include their combination with other antitumor therapies, looking for a synergistic or additive effect by using drugs with non-overlapping toxicities and mechanisms of action.

For example, preclinical studies show that the treatment of OS cells with sorafenib results in the upregulation of mTORC2 as an escape mechanism, which sets a rationale for the coadministration of mTOR inhibitors [124]. A non-randomized phase II clinical trial evaluated the combination of sorafenib and everolimus in 38 patients with relapsed OS, showing a PFS rate at 6 months of 45% and a discrete increase in toxicity when compared to sorafenib in monotherapy [81].

The combination of imatinib plus everolimus has been studied in a phase II clinical trial in 43 patients with advanced chordoma, achieving nine PR (20.9%), 24 SD (55.8%), and a median PFS of 11.5 months according to the Choi criteria. Three PR were observed among the 13 patients pre-treated with imatinib in monotherapy. Interestingly, responsive patients had a higher proportion of cells with the phosphorylation of S6/4EBP1 as a surrogate of mTOR activity [89]. Similarly, a retrospective study on 10 advanced imatinib-resistant chordomas (one with a clival chordoma) treated with imatinib plus sirolimus reported a clinical benefit of 89%, with seven PR and one SD, according to the Choi criteria [125].

The coadministration of TKIs and conventional chemotherapy has also been explored. A phase I/II clinical trial combining lenvatinib with etoposide plus ifosfamide in 35 patients with relapsed OS found a PFS of 51% at 4 months [83]. A phase I trial evaluated regorafenib in combination with vincristine and irinotecan in pediatric patients with recurrent or refractory solid tumors including 12 ES, with discrete clinical activity (two PR) [126]. The phase I trials of gefitinib with irinotecan [127] and erlotinib with temozolomide [128] suggest that these are tolerable but scarcely effective regimes against recurrent OS.

Pharmacokinetics and overlapping toxicity should be considered when trying these combinations. Recent studies have shown an exacerbated cardiotoxicity in patients treated with anthracyclines and sorafenib [129] as well as an increased toxicity of irinotecan when combined with regorafenib, possibly due to the inhibition of UGTA1A126.

There is also an interesting rationale for the combination of TKIs with immunotherapy due to their modulating effects on the tumor microenvironment. A growing body of data from preclinical studies show that multi-target TKIs, especially cabozantinib and lenvatinib, decrease the number of myeloid-derived suppressor cells (MDSC) and tumor-associated macrophages (TAMs), and increase the tumor infiltration of dendritic cells, NK cells, and CD8+ lymphocytes [130].

A single-arm phase II trial of apatinib plus anti-PD1 camrelizumab (APFAO) in 43 patients with OS showed an ORR of 20.9% and a 6-month PFS of 50.9%, with a statistically longer PFS in patients with a programmed cell death 1 ligand-1 (PD-L1) tumor proportion score ≥ 5% (*p* = 0.004) [86]. A phase I/II trial of sunitinib and nivolumab in advanced sarcomas (IMMUNOSARC) included a cohort of 40 pre-treated patients with bone sarcomas (17 OS, 14 CS, eight ES, one UPS) reaching a 6 month PFS > 30% and a median OS of 14.2 months [85]. Axitinib plus pembrolizumab has shown preliminary activity in 36 patients with advanced sarcomas in a phase II clinical trial, though just two bone sarcomas (1 ES, 1 CS) were included [131]. Some retrospective data suggest a modest antitumor activity of pazopanib plus nivolumab (three OS with one PR and one SD) [132] and pazopanib plus pembrolizumab (one UPS with disease regression for 10 months) [133].

## 4. Conclusions

Tyrosine kinase receptors are key for the activation of cellular downstream pathways that contribute to the pathogenesis of bone sarcomas. TKIs are progressively emerging as a therapeutic alternative for patients with recurrent unresectable or metastatic diseases that can no longer benefit from surgery or conventional chemotherapy.

Regorafenib and cabozantinib are the most represented agents in phase II clinical trials, especially in osteosarcoma, though apatinib may have a higher rate of objective responses. Their effectiveness in Ewing sarcoma seems lower, probably due to quicker development of acquired resistance. Axitinib, sunitinib, sorafenib, and lenvatinib also seem to have antitumor activity, especially in osteosarcoma patients. Dasatinib has shown modest efficacy in chondrosarcoma, whereas imatinib—and lapatinib when EGFR overexpression is detected—may be useful in advanced chordoma.

Although a significant proportion of patients achieve temporary disease stabilization, maintained responses are anecdotal with TKIs in monotherapy. Some studies have begun to explore the combination of TKI with chemotherapy, other targeted therapies, and immune-checkpoint inhibitors in an attempt to obtain a synergistic effect without significantly increasing toxicity.

Further research is required to properly understand the mechanisms of resistance, find biomarkers predictive of response, explore the benefit of sequentially using several TKIs in responsive patients, and understand how to optimize their use in clinical practice, both alone and in combination with other systemic therapies.

## Figures and Tables

**Figure 1 ijms-23-13784-f001:**
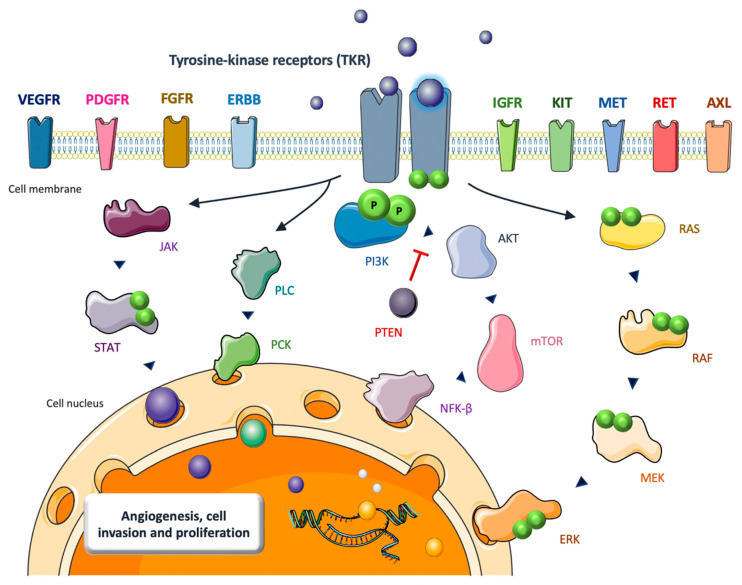
Main downstream oncogenic pathways unleashed by ligand-interaction and dimerization of the transmembrane tyrosine-kinase receptors (TKR) with a potential relevance as therapeutic targets in primary bone tumors: JAK/STAT, PLC/PCK, PI3K/Akt/mTOR, and RAS/RAF/MEK/ERK (MAPK).

**Table 1 ijms-23-13784-t001:** Phase II trials with TKIs in bone sarcomas. OS: osteosarcoma; ES: Ewing sarcoma; CI: confidence interval; mPFS: median progression-free survival; OS: overall survival; PR: partial response; SD: stable disease; DCR: disease control rate (PR + SD); ORR: overall response rate; w: weeks; m: months; HR: hazard ratio; AST: aspartate aminotransferase; ALT: alanine aminotransferase; ALP: alkaline phosphatase; LDH: lactate dehydrogenase; PPS: palmar-plantar syndrome.

Clinical Trial (Phase)	Agent	Tumor	*N*	Age Range (Years)	Outcomes		G3/G4 Adverse Events
Duffaud et al. (REGOBONE) (phase II multi-cohort)	Regorafenib (vs. placebo)	OS [75]	38	21–50	DCR: 17/26 (65%; 95% CI 47–95) (vs. 0% in placebo); mPFS: 16 w (95% CI 8.0–27.3) vs. 4 w (95% CI 3.0–5.7)		Hypertension (24%), PPS (10%), fatigue (10%), hypophosphatemia (10%), chest pain (10%)
		ES [76]	41	16–59	PR: 5/36 (21.7%) (vs. 0% in placebo); mPFS: 11.4 w (95% CI 4.6–22.9) vs. 3.9 w (95% CI 3.3–7.3)		Diarrhea (13%), PPS (13%), thrombocytopenia (9%), fatigue (9%), mucositis (9%), febrile neutropenia (9%)
Davis et al. (SARC024) (phase II) [77]	Regorafenib (vs. placebo)	OS	42	18–76	mPFS: 3.6 m (95% CI 2.0–7.6) vs. 1.7 m (95% CI 1.2–1.8) (HR 0.42; 95% CI 0.21–0.85; *p* = 0.017)		Hypertension (14%), rash (9%), hypophosphatemia (9%), extremity pain (9%), thrombocytopenia (9%), PPS (5%)
Attia et al. (REGO) (phase II) [78]	Regorafenib	ES	30	19–65	PR: 3/28 (10.7%); SD: 18/28 (64.3%); mPFS: 3.6 m (95% CI 2.8–3.8)		Hypophosphatemia (20%), hypertension (6.7%), ALT increase (6.7%), fatigue (3.3%), abdominal pain (3.3%), diarrhea (3.3%), hypokalemia (3.3%), oral mucositis (3.3%), neutropenia (3.3%), rash (3.3%)
Italiano et al. (CABONE) (phase II) [79]	Cabozantinib	OS/ES	90	20–53	OS (*n*: 42)	PR: 5/42 (12%; 95% CI 4–26), SD: 14/42 (33%; 95% CI 20–50); mPFS: 7.2 m PR (95% CI 4.7–10.9); 4.5 m SD (95% CI 1.8–9.5); 1.8 m PD (95% CI 0.8–1.9)	Hypophosphatemia (9%), neutropenia (7%), AST increase (6%), PPS (6%), pneumothorax (6%)
					ES (*n*: 39)	PR: 10/39 (26%; 95% CI 13–42); mPFS: 4.4 m (95% CI 3.7–5.6)	
Grignani et al. (phase II) [80]	Sorafenib	OS	35	15–62	ORR: 14% (95% CI 2–26) DCR: 49% (95% CI 31–67) mPFS: 4 m (95% CI 2–5)		PPS (9%), anemia (6%), thrombocytopenia (6%), CK elevation (6%), leucopenia (3%), rash (3%), mucositis (3%), nausea (3%), fatigue (3%), lipase elevation (3%), pneumothorax (3%), bleeding (3%)
Grignani et al. (phase II) [81]	Sorafenib + everolimus	OS	38	18–64	ORR: 10% (95% CI 0.3–21); DCR: 63% PFS at 6 m: 45% (95% CI 28–61)		Hypophosphatemia (16%), lymphopenia (16%), PPS (13%), thrombocytopenia (11%), fatigue (5%), mucositis (5%), diarrhea (5%), anemia (5%), pneumothorax (3%)
Gaspar et al. (phase I/II) [82]	Lenvatinib	OS	31	9–22	ORR: 7% (95% CI 0.8–22.1) PFS at 4 m: 29% (95% CI 14–48); mPFS: 3 m (95% CI 1.8–5.4)		Hypertension (3%), diarrhea (3%), proteinuria (3%), decreased weight (3%), abdominal pain (3%)
Gaspar et al. (phase I/II) [83]	Lenvatinib + IF/VP16	OS	35	2–25	PFS at 4 m: 51% (95% CI 34–69);		Neutropenia (77%), thrombocytopenia (71%), anemia (54%), leukopenia (54%)
Chugh et al. (phase II) [84]	Imatinib	OS/ES/ others	185	14–83	OS (*n*: 27)	PR: 0/27 (0%); SD: 5/27 (19%)	No G3/G4 adverse events reported
					ES (*n*: 13)	PR/SD: 0/13 (0%)	
Palmerini et al. (IMMUNO-SARC) (phase I/II) [85]	Sunitinib + nivolumab	OS/ES/ others	40	21–74	DCR: 24/40 (60%) (1 CR, 1 PR, 22 SD) mPFS 3.7 m (95% CI 3.4–4) mOS 14.2 m (95% CI 7.1–21.3)		Neutropenia (10%), anemia (10%), ALT/AST increase (7.5%), fatigue (5%), oral mucositis (5%), thrombocytopenia (2.5%), dysphagia (2.5%), gastric hemorrhage (2.5%), malaise (2.5%), thromboembolism (2.5%), pneumonitis (2.5%)
Xie et al. (APFAO) (phase II) [86]	Apatinib + camrelizumab	OS	43	11–43	ORR: 9/43 (20.9%) PFS at 6 m: 50.9% (95% CI 34.6–65.0)		Wound dehiscence (14%), ALP increase (9.3%), AST/ALT increase (9.3%), blood bilirubin increase (9.3%), hypertriglyceridemia (7.0%), anorexia (7.0%), weight loss (7.0%), pneumothorax (7.0%), platelet count decrease (4.7%), diarrhea (4.7%), PPS (4.7%), limb pain (4.7%), leukopenia (4.7%), rash (4.7%), oral mucositis (4.7%), hypertension (4.7%), toothache (4.7%), nausea (4.7%), non-cardiac chest pain (4.7%), hypothyroidism (2.3%), blood LDH increase (2.3%), proteinuria (2.3%), cough (2.3%), hemorrhoidal hemorrhage (2.3%), fatigue (2.3%), peripheral neuroinflammation (2.3%)
Schuetze et al. (phase II) [87]	Dasatinib	CS	33	22–87	ORR: 6/33 (18.2%); mPFS: 5.5 m		Pain (17%), dyspnea (11%), pleural effusion (6%), diarrhea (5%), anemia (3%), thrombocytopenia (2%), neutropenia (<1%), lymphopenia (<1%)
		Chordoma	32		ORR: 6/32 (18.8%); mPFS: 6.3 m		
Stacchiotti et al. (phase II) [88]	Imatinib	Chordoma	50	24–86	PR: 1/50 (2%); SD 35/50 (70%); ORR: 2% (95% CI 0–5.3); mPFS: 9.2 m		Fluid retention (29%)
Stacchiotti et al. (phase II) [89]	Imatinib + everolimus	Chordoma	40	49–70	PR: 9/40 (20.9%); SD 24/40 (55.8%) mPFS: 11.5 m (95% CI 4.6–17.6)		Infection (16%), fatigue (9%), anemia (2%), leukopenia (2%), febrile neutropenia (2%), thrombocytopenia (2%), cardiac ischemia (2%)
Stacchiotti et al. (phase II) [90]	Lapatinib	Chordoma (EGFR)	18	35–75	PR: 6/18 (33.3%); SD: 7/18 (38.9%) mPFS: 6 m (95% CI 3–8)		Anemia (5.6%), rash (5.6%), thromboembolism (5.6%)

## Data Availability

Not applicable.

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
