# Peer review of "Receptor Tyrosine Kinase Inhibitors for the Treatment of Recurrent and Unresectable Bone Sarcomas"

_ijms, 2022, doi:10.3390/ijms232213784_

Round 1
Reviewer 1 Report
Albarran and co-authors have assembled a comprehensive and readable review of the current status of tyrosine kinase inhibitor (TKI) therapy in bone sarcomas, based largely on the results of ongoing and completed clinical studies. As the authors note, several other reviews with a similar focus have been published recently (for example Tian, Niu and Yao, osteosarcoma, Frontiers in Oncology, 2020; Just, van Mater and Wagner, osteosarcoma and Ewing sarcoma, Pediatric Blood and Cancer, 2021 and Kyriazoglou et al, sarcoma, Oncology Letters, 2022), however, this work is broader and organised with a focus on disease type rather than TKI.
I would recommend publication after the authors have addressed the following minor issues.
Is there a missing reference for the diagnosis and treatment guidelines in the final paragraph of Introduction section 1a?
Please provide a reference (even if just to the website) for the term RECIST criteria.
In section 2b, I think the reference to the FGFR1 N546 mutation should be FGFR1 N546K?
I wonder why Table 1 was presented as supplementary material rather than part of the main text? The following abbreviations are missing from the table 1 footnote: CI and LDH. I would also have appreciated a second table listing the various TKIs mentioned in the article alongside their (multiple) kinase targets.
I would like to suggest some minor language changes which could improve reader understanding as follows:
Abstract, line 22: replace the word 'weaker' with 'less advanced'
Introduction, section 1b: consider the following replacement for the final phrase in paragraph 5 (line 09-91) 'low response rates such that the prognosis of recurrent ES remains very poor'
Section 2a - chromothripsis is mis-spelt - this shows correct spelling. Also consider this replacement for the rest of the sentence: 'a single genetic catastrophe (a phenomenon known as chromothripsis), whereas in other solid tumors only 2-3% show such features.'
line 129: 'differentiation of mesenchymal stem cells'
line 145: 'phosphorylate tyrosine residues in specific substrates.'
line 173: 'The VEGFR family is a common target [...], some of which show specificity for certain isoforms. Imatinib is the most powerful inhibitor of all three isoforms.'
line 195: ' some evidence implicating the PDGF/PDGFR pathway in...'
line 210: 'Though less investigated to date, other TKRs such as [....] are overexpressed and implicated in the progression of OS, and are associated with a poorer prognosis and'
Section 2b, first sentence, should be 'The key event' (not clue event)
line 221; I think this should be 'proteins that favor the dissemination of aberrant cells'
line 243 onwards: 'As observed in OS, FGFR is highly expressed in ES cells. In particular, FGFR1 is highly activated [...] EWS/FLI1 fusion protein can significantly reduce the expression of SPRY1, a negative regulartor of the FGFR-activated MAPK pathway'
line 259: I think this should read 'HER3 and HER4 is increased in ES cell lines and is associated with shorter survival times'?
Section 2c, first sentence ' OS and ES, research has also begun'
line 284: 'clinical samples, and showed that several TKIs suppressed S6 phosphorylation'
Section 3a: line 392 'pazopanib is a multi-TKI that targets VEGFR1/2/3, PDGFR and FGFR, and is more widely used in soft tissue sarcomas.'
Section 3b: line 426 'both markers (3+/4+), giving rise to the hypothesis'
Section 3c, first sentence: 'and the consequent rarity of unresectable and disseminated status'
Section 3d line 494: 'This suggests that bone sarcomas have either'
line 501: 'OS patients from the CABONE trial with'
Section 4, first sentence 'cellular downstream pathways'
Figure 1 legend: 'dimerization of transmembrane tyrosine-kinase' and PI3K/Akt (capital K in PI3K)
Throughout: 'in contrast with' should be 'in contrast to'
With thanks to the authors
Reviewer 2 Report
A needed review. From the prospective of patients and clinicians these agents are active; although CR and PR are uncommon, SD is common. Thus, the concept of disease control rate (DCR=PR+CR) is an important message and needs to be MUCH more evident in the text of the paper as well as in table 1. For example when you look at waterfall plot in ES of cabozantinib PR+SD is an excellent proportion. Once this is seen throughout the manuscript, this will show better and more hopeful perspective - from one that these agents seem to be without activity to one in which TKI can provide some hope for temporary disease control (PR+ SD).
Author Response
Thank you so much for your positive review. The importance of disease stabilization with TKIs in monotherapy has been explained with detail throughout the text and has been highlighted in the conclussion of the revised manuscript, as well as the need of combining them with other therapies to reach higher rates of objective responses.